# Comparison of Resveratrol Supplementation and Energy Restriction Effects on Sympathetic Nervous System Activity and Vascular Reactivity: A Randomized Clinical Trial

**DOI:** 10.3390/molecules26113168

**Published:** 2021-05-26

**Authors:** Gustavo Henrique Ferreira Gonçalinho, Alessandra Roggerio, Marisa Fernandes da Silva Goes, Solange Desirée Avakian, Dalila Pinheiro Leal, Célia Maria Cassaro Strunz, Antonio de Padua Mansur

**Affiliations:** Chronic Coronariopathies Clinical Unit, Heart Institute, School of Medicine, University of São Paulo, São Paulo 05403-000, SP, Brazil; ghfg93@gmail.com (G.H.F.G.); alessandra.roggerio@incor.usp.br (A.R.); marisa.goes@incor.usp.br (M.F.d.S.G.); solange.avakian@incor.usp.br (S.D.A.); dalila.pinheiro.leal@hotmail.com (D.P.L.); labcelia@incor.usp.br (C.M.C.S.)

**Keywords:** noradrenaline, endothelial dysfunction, cardiovascular diseases, resveratrol, calorie restriction, sympathetic nervous system, vascular reactivity

## Abstract

*Background:* Chronic sympathetic nervous system activation is associated with endothelial dysfunction and cardiometabolic disease, which may be modulated by resveratrol (RSV) and energy restriction (ER). This study aimed to examine the effects of RSV and ER on plasma noradrenaline (NA), flow-mediated vasodilation (ed-FMD), and endothelium-independent nitrate-mediated vasodilation (ei-NMD). *Methods*: The study included 48 healthy adults randomized to 30-days intervention of RSV or ER. *Results:* Waist circumference, total cholesterol, HDL-c, LDL-c, apoA-I, and plasma NA decreased in the ER group, whilst RSV increased apoB and total cholesterol, without changing plasma NA. No effects on vascular reactivity were observed in both groups. Plasma NA change was positively correlated with total cholesterol (*r* = 0.443; *p* = 0.002), triglycerides (*r* = 0.438; *p* = 0.002), apoA-I (*r* = 0.467; *p* = 0.001), apoB (*r* = 0.318; *p* = 0.032) changes, and ei-NMD (OR = 1.294; 95%CI: 1.021–1.640). *Conclusions*: RSV does not improve cardiometabolic risk factors, sympathetic activity, and endothelial function. ER decreases plasma NA and waist circumference as well as improves blood lipids, but does not modify endothelial function. Finally, plasma NA was associated with ei-NMD, which could be attributed to a higher response to nitrate in patients with greater resting sympathetic vasoconstriction.

## 1. Introduction

The endothelium is a key regulator of vascular homeostasis, acting not just as a barrier, but as an active signal transducer for circulating influences that modify the vessel wall phenotype. The loss of endothelial function is an initial step of atherosclerosis and participates in lesion development and later clinical complications [1,2]. Although there are several techniques for evaluating the endothelial function, non-invasive techniques have proven useful, both for their clinical applicability and their predictive feature for future cardiovascular disease [3]. The brachial artery ultrasound technique using hyperemia to induce endothelium-dependent flow-mediated dilation (ed-FMD) and endothelium-independent vasodilation assessment are inversely associated with cardiovascular risk [4,5]. It has also been observed that cardiovascular risk factors such as body mass index (BMI), smoking, and blood lipids are important determinants of vascular reactivity [6].

It has been shown that vascular reactivity is regulated by sympathetic nervous system (SNS) activation [7]. The SNS modulates vasoconstriction and vasorelaxation through alpha_1_ and beta_2_-adrenoreceptors (AR) signaling, respectively. Changes in beta-adrenergic signaling are observed in aging, heart failure, metabolic syndrome, hypertension, obesity, and diabetes [8,9]. Under such conditions, defective beta_2_-AR-signaling of blood vessels leads to a progressive shift toward vasoconstriction and increased systemic noradrenaline (NA) release [8,9]. Because NA is the main SNS neurotransmitter, SNS activity may be measured by plasma NA concentration [10]. Previous works have shown that plasma NA is elevated in advanced age, obesity, heart failure, and essential hypertension [11,12,13,14]. Furthermore, plasma NA is a predictor of cardiovascular events [13,15,16,17].

Because SNS overdrive has been suggested as a crucial etiology for developing a variety of cardiovascular diseases, pharmacological and non-pharmacological sympatholytic methods have been widely adopted and recommended to treat these conditions [18]. Among the non-pharmacological methods, the diet has an important role in modulating SNS activity. It has been demonstrated that energy restriction (ER), besides improving metabolic parameters, reduces NA mainly in obese patients [19]. This indicates that the reduction of SNS activity plays an important role in weight loss [7,19].

Although energy restriction presents numerous metabolic benefits, long-term adherence becomes increasingly difficult. Therefore, substances called energy restriction mimetics that have mechanisms similar to a hypocaloric diet and beneficial cardiometabolic effects have been studied. Resveratrol (RSV), the main representative of this group of substances, is a non-flavonoid polyphenolic compound derived from stilbene and is also a phytoalexin present in some foods such as peanuts, berries, and grapes [20]. RSV supplementation has been associated with improvement of cardiometabolic parameters such as blood pressure, plasma glucose, triglycerides, and LDL-c reduction, and augmentation of plasma HDL-c and insulin sensitivity [20]. Due to its antioxidant properties, it has been suggested that RSV affects redox balance and inflammation signaling, modulating positively nitric oxide (NO) production and consequently vascular function [20]. In addition, NO is a messenger in the central nervous system and has an important influence on the rostral ventrolateral medulla (RVLM) and sympathetic nerve activity. RVLM plays a pivotal role in the regulation of vascular tone and maintenance of blood pressure (BP), and its stimulation increases BP and sympathetic outflow. Therefore, the increased NO production promoted by RSV lead to RVLM activity reduction and decreased sympathetic outflow, BP, and heart rate [21]. However, clinical trials investigating the effects of RSV on SNS activity are scarce.

Therefore, this study aims to compare the effects of RSV supplementation and ER diet on vascular reactivity and SNS activity measured by plasma NA in healthy men and women.

## 2. Results

### 2.1. Baseline and Post-Intervention Data

Baseline and post-intervention characteristics of the participants are summarized in Table 1. Sample overall age was 58.5 (SD: 3.5) years, without differences among groups. There were no statistically significant differences in the baseline characteristics among groups, and all subjects adhered to both interventions. At the end of the study, RSV supplementation increased total cholesterol (9.88 mg/dL; *p* = 0.030) and apoB (0.05 g/L; *p* = 0.034) without statistically significant differences in the other parameters. On the other hand, ER decreased waist circumference (−2.5 cm; *p* = 0.011) but without a statistically significant difference in BMI (*p* = 0.083). Additionally, a reduction of serum total cholesterol (−13.54 mg/dL; *p* = 0.007), HDL-c (−3.33 mg/dL; *p* = 0.008), LDL-c (−8.38 mg/dL; *p* = 0.031), apoA-I (−0.10 g/L; *p* = 0.011) and plasma NA (−93.86 pg/dL; *p* = 0.008) was observed. Furthermore, differences between the mean changes of serum total cholesterol (RSV: 9.88 mg/dL vs. ER: −13.54 mg/dL; *p* < 0.001), LDL-c (RSV: 7.21 mg/dL vs. ER: −8.38 mg/dL; *p* = 0.006), triglycerides (RSV: 24.58 mg/dL vs. ER: −9.21 mg/dL; *p* = 0.048), apoB (RSV: 0.06 g/L vs. ER: −0.06 g/L; *p* = 0.004), and glucose (RSV: 2.54 mg/dL vs. ER: −3.08 mg/dL; *p* = 0.033) were observed among the RSV and ER groups. No statistically significant differences were observed in vascular reactivity parameters in both groups.

### 2.2. Correlations between Changes (Δ = Post-Intervention—Baseline) of Vascular Reactivity, Plasma Noradrenaline, and Cardiometabolic Parameters

Correlations of changes in vascular reactivity and plasma NA with cardiometabolic parameters are summarized in Table 2. In terms of vascular reactivity, overall plasma NA changes was positively correlated with ei-NMD changes (*r* = 0.338; *p* = 0.041), but not with ed-FMD changes (*r* = 0.129; *p* = 0.446). Overall plasma NA changes were also positively correlated with total cholesterol (*r* = 0.443; *p* = 0.002), triglycerides (*r* = 0.438; *p* = 0.002), apoA-I (*r* = 0.467; *p* = 0.001), and apoB (*r* = 0.318; *p* = 0.032) changes. In the RSV group, plasma NA changes correlated with waist circumference (*r* = 0.514; *p* = 0.020), triglycerides (*r* = 0.494; *p* = 0.014), apoA-I (*r* = 0.508; *p* = 0.011), and glucose (*r* = 0.420; *p* = 0.041) changes, whereas, in the ER group, these correlated with total cholesterol (*r* = 0.449; *p* = 0.036) and NEFA (*r* = −0.455; *r* = 0.033) changes. ed-FMD changes were associated only with waist circumference change in the ER group (*r* = −0.536; *p* = 0.033), whereas ei-NMD changes were correlated with apoA-I (*r* = 0.469; *p* = 0.043) and HOMA-IR (*r* = 0.561; *p* = 0.012) changes in the RSV group.

### 2.3. Cardiometabolic Variables and Plasma Noradrenaline Changes

We applied linear regression analyses to further evaluate the contributions of cardiometabolic health variables in vascular reactivity and plasma NA changes. Total cholesterol and triglycerides were predictors of plasma NA changes, where for each unit increase of total cholesterol and triglycerides, NA increased 2.61 pg/dL (95%CI: 1.002–4.211; R2 = 0.196) and 0.94 pg/dL (95%CI: 0.258–1.619; R2 = 0.149), respectively. NA was a predictor of ei-NMD changes, where for each unit increase of NA, ei-NMD increased 0.02 % (95%CI: 0.001–0.039; R2 = 0.114).

Simple and multivariate logistic regressions were applied to verify the association between NA and ei-NMD changes (Table 3). It was found that ei-NMD, triglycerides, and total cholesterol were positively associated with NA, but only ei-NMD presented an independent association (β = 0.258; OR = 1.294; 95%CI: 1.021–1.640).

## 3. Discussion

The present study evaluated the effects of ER and RSV on sympathetic nervous system activity, vascular reactivity, and cardiovascular risk factors in healthy men and women after 30 days of intervention. The main results showed that ER but not RSV supplementation decreased plasma NA, and that both interventions did not change endothelium-dependent and -independent vasodilation.

There is substantial evidence in support of the SNS being exceedingly active in individuals with cardiometabolic diseases and its key metabolic alterations, central obesity, and insulin resistance, and that plasma noradrenaline is increased in these conditions [10,19]. Adiposity is an important modulator of SNS activity. In a study evaluating nonobese individuals, diet-induced modest weight gain elicited SNS activation, which was correlated with blood pressure rise [22]. Plasma NA concentrations also increased following weight gain in longitudinal studies, and the rise of this neurotransmitter preceded increases in blood pressure and plasma leptin [23,24]. The present study showed that ER significantly reduced plasma NA, meaning a decrease in SNS activity. This result is in line with previous works that demonstrated the sympathoinhibitory effect of ER and weight loss [25,26,27]. This may be attributed to the observed waist circumference reduction. Visceral adiposity, indirectly measured by waist circumference, in addition to being associated with features of metabolic syndrome, is associated with higher sympathetic neural activation [28,29]. White adipose tissue contributes toward SNS activity through leptin and non-esterified fatty acids (NEFA) [19]. Leptin impairs glucose transporter-4 (GLUT-4) translocation in skeletal muscle and induces hyperinsulinemia, which results in coactivation of the SNS [19]. Moreover, leptin acts centrally in the hypothalamus and brainstem, which control multiple metabolic functions via melanocortin-system-dependent pathways to increase sympathetic activity [19]. However, leptin was not measured in the present study. Circulating plasma NEFA is positively correlated with obesity and insulin resistance [19]. Like leptin, NEFA can act locally in peripheral tissue to disrupt insulin signaling and impair glucose uptake or can act centrally [19]. However, there was no significant difference in NEFA at the end of the study in both groups. Although the 30-day ER intervention reduced waist circumference, it was not sufficient to bring about a reduction in BMI. This result is in line with a previous work that showed no significant BMI reduction in four weeks, but only after 16 weeks with ER alone [25].

The decrease in plasma NA caused by ER may also be attributed to serum total cholesterol, LDL-c, and apoB reduction. Moreover, total cholesterol, apoB, apoA-I, and triglycerides were positively correlated with plasma NA in both intervention groups of the present study. It was shown in previous studies that acute ER causes cholesterol synthesis inhibition independently of weight loss, which leads to a reduction in plasma cholesterol and apoB-associated lipoproteins [30,31]. Furthermore, 3-hydroxy-3-methyl-glutaryl-coenzyme A (HMG-CoA) reductase inhibitors have demonstrated sympatholytic effects in previous randomized controlled trials, highlighting the relationship between sympathetic activity and cholesterol metabolism [32,33]. The mechanism behind the link between cholesterol and plasma NA reduction in the present study may be due to atherosclerosis inhibition within the aortic arch and carotid arteries, which might improve the sensitivity of the high-pressure baroreceptors, leading to an inhibition of SNS [33]. The triglyceride reduction within the ER group was not statistically significant. This was an unexpected result because ER reduces blood triglyceride content [30,31]. However, in regression analyses, triglycerides were significant predictors of plasma NA changes. This is in line with the findings of a meta-analysis that have demonstrated a direct association between triglycerides and SNS activation [34]. One possible mechanism is that triglycerides can cross the blood–brain barrier and induce central leptin and insulin receptor resistance [35]. The hormone secretion increases due to central leptin resistance, which is selective, causing an augmented sympathetic activity [19].

RSV supplementation did not reduce plasma NA significantly. Prior studies have linked RSV and SNS activity. It was found that RSV inhibits catecholamine synthesis and secretion in cultured bovine adrenal medullary cells by suppression of Na^+^ and Ca^2+^ influx through nicotinic acetylcholine receptor-ion channels and voltage-dependent Ca^2+^ channels, which evokes synthesis and secretion of catecholamines [36]. However, lower RSV concentrations might increase catecholamine synthesis due to the activation of plasma membrane estrogen receptors [36]. Other works have shown that RSV reduces heart NA levels, which is associated with post-infarction arrhythmia and cardiomyocyte hypertrophy led by sympathetic overdrive [37,38]. Although these findings support the sympathoinhibitory effects of RSV, there is a lack of clinical evidence, and the present study does not corroborate this effect in SNS activity. As far as it is known, this is the first clinical study that assessed RSV effects on SNS activity. One limitation is that the use of plasma NA as a marker of SNS activity is still controversial, not necessarily reflecting muscle sympathetic nerve activation, which is a better parameter for this purpose [10,34].

Unlike what was expected, RSV did not improve blood lipids; in contrast, total cholesterol and apoB were higher at the end of the study. Scientific literature suggests that RSV improves lipid metabolism [20,39,40,41,42]. Preclinical evidence showed that RSV decreased blood triglycerides and LDL-c levels, and increased HDL-c levels [39]. The plausible mechanisms of serum cholesterol improvement are HMG-CoA reductase inhibition and the increase of LDL receptors in hepatocytes, thereby contributing to the enhancement of LDL particle clearance [20]. Due to the antioxidant properties of RSV, protective effects on LDL and HDL oxidation have been observed [42]. HDL oxidation alters apoA-I and paraoxonase-1 functions and jeopardizes reverse cholesterol transport. It was previously demonstrated that RSV prevented HDL oxidation, increased ABCA-1-dependent cholesterol efflux, and decreased macrophage cholesterol influx, in addition to the reduction of LDL oxidation [42]. These findings support the ability of RSV to decrease serum total cholesterol and LDL-c and increase HDL-c. However, clinical evidence remains controversial. Findings of a meta-analysis of randomized clinical trials indicated RSV could not change plasma triglycerides, LDL-c, and HDL-c concentrations, but decreased total cholesterol [40]. Another meta-analysis assessing the effects of RSV in blood lipids in patients with type 2 diabetes mellitus showed that shorter time interventions (<6 months) led to an increase in LDL-c and total cholesterol, which is in line with the findings of the present study [41]. This result remains inconclusive and needs further explanations in future research.

With regard to triglycerides, shorter time interventions (<6 months) increased plasma concentrations, which is also in line with the present study, despite the result not being statistically significant [41]. On the other hand, studies with longer intervention time have shown a triglyceride-lowering effect, which may be attributable to the activation of AMP-activated protein kinase (AMPK) and NAD-dependent protein deacetylase sirtuin-1 (SIRT1), which in turn inhibit sterol regulatory element-binding protein 1 activity and increases fatty acids oxidation, respectively [41]. A previous work of our group demonstrated that RSV increased SIRT1 gene expression, despite triglyceride reduction was not observed [43]. The reason for the findings with regard to lipid profile in the present study may be attributed to the short-time RSV supplementation. It is important to note that serum triglycerides were positively correlated with NA in the RSV group and were predictors of plasma NA change, which may indicate that triglyceride reduction in response to RSV supplementation shown in previous studies is linked with sympathoinhibitory effects [34]. Further investigations are necessary to clarify the effects of long-term RSV supplementation on triglycerides and their relationship with SNS activity and leptin.

Both RSV supplementation and ER did not change serum hs-CRP, glucose, and HOMA-IR significantly at the end of the study. There is evidence that 1000 mg/day of RSV may reduce glycemia [44]. However, it seems that beneficial effects of RSV on glucose metabolism occur when administration dose is higher than 500 mg and in a period higher than 10 weeks, explaining why in the present 30-day intervention study there were no significant alterations observed [45]. Furthermore, the lack of effect of RSV on inflammatory biomarkers corroborates the results of a previous meta-analysis [46]. The period of the study was also not sufficient to cause significant weight loss by the ER, making it not possible to observe significant changes in glucose metabolism. The lack of effect of ER on inflammatory biomarkers also corroborates the results of a previous study [47].

A previous study demonstrated that plasma NA was an independent predictor of ed-FMD in healthy adults, having an inverse association attributed to the increase in SNS activity related to the aging process [48]. Additionally, an independent and positive association was shown between plasma NA and ei-NMD, in line with the results of the present study. Long-term neurohormonal activation by excessive plasma NA alters vascular wall structure and stiffness, leading to a vasoconstrictive state [49]. Previous reports showed that excess NA impairs ei-NMD in pheochromocytoma patients, exhibiting blunted smooth muscle cell NO response to exogenous NO [50]. However, in the present study, independent association of ei-NMD and plasma NA may be attributed to a greater response to vasoactive agents, found in patients with greater resting vasoconstriction. Cyclic guanosine monophosphate (cGMP) and NO act as endogenous inhibitors of NA release in vascular smooth muscle. Excess NA may increase endothelial cGMP and NO activation, which could lead to a higher relaxation response of vascular smooth muscle cells to endothelial NO mediated by isosorbide dinitrate (ei-NMD) found in the present study [51].

HOMA-IR change was positively correlated with ei-NMD at the RSV group, despite no statistically significant changes being observed at the end of the study. As above-mentioned, ei-NMD may be increased due to greater resting vasoconstriction in healthy adults, which reduces glucose uptake on skeletal muscle cells [10,19].

In the present study, RSV supplementation did not change endothelium-dependent either endothelium-independent vasodilation parameters. These are in line with a previous meta-analysis that showed that ed-FMD response to RSV occurs in interventions with a duration higher than eight weeks [52]. Furthermore, RSV did not change ed-FMD in overweight patients [52].

ER decreased plasma NA in the present study, but no changes in vascular reactivity remained significant. In a randomized controlled trial, it was observed that weight loss did not change ed-FMD, despite improvement in cardiometabolic parameters [53]. On the other hand, another study showed that ed-FMD improved due to ER in the first week of follow-up. However, the study included obese women [54]. In addition, the intervention was a very low-calorie diet, which implied 18 dropouts during the study. Furthermore, ed-FMD improvement was associated with BMI reduction [54]. The explanation of why ed-FMD did not improve with ER in the present study is because BMI reduction was not observed, as found in a previous study [55]. This was probably due to the short follow-up period of the study. Additionally, ed-FMD improvement was more pronounced in patients with BMI higher than 35 and with comorbidities, and subjects of our study were slightly overweight [55]. The absence of changes in blood flow to RSV and ER interventions may also be related to a small number of subjects because sample size calculation used changes in NA plasma levels, instead of changes in FMD. Nevertheless, we hypothesized that the amount of 100 pg/dL (SD = 100 pg/dL) changes in NA plasma levels would be associated with an increase of 40% in FMD, as shown in previous studies [52,53]. The homogeneous group of healthy subjects may also explain the lack of changes in FMD after the interventions. However, we did find a moderate correlation between changes in NA plasma levels and ed-FMD. Therefore, this latter result may be more important than the comparison between pre- and post-intervention data.

The strength of the study was the use of a well-established marker of endothelial dysfunction, which is a predictor of cardiovascular disease events. However, plasma biomarkers of endothelial dysfunction were not assessed. Other limitations were possible occasional non-adherence to ER, the slight but not significant differences in baseline NA plasma concentrations, and in the short follow-up, which made it difficult to observe more expressive effects of RSV supplementation and ER on vascular reactivity. The limitations around the use of plasma NA were: (1) only a small fraction of NA diffused into plasma where it was measured; (2) plasma NA concentration was dependent on clearance rate, and not only sympathetic tonus and NA secretion; and (3) the sources of plasma NA were not identified, though regional sympathetic responses cannot be measured [10].

In conclusion, the results of the study showed that short-term RSV supplementation does not improve cardiometabolic risk factors, sympathetic activity, and endothelial function, whereas short-term ER decreases plasma NA and waist circumference as well as improves blood lipids, but does not modify the endothelial function in overweight healthy adults. Furthermore, plasma NA change was independently associated with ei-NMD, which may be attributed to greater resting vasoconstriction caused by increased sympathetic activity.

## 4. Materials and Methods

### 4.1. Participants and Study Design

A randomized controlled trial was conducted in 48 individuals (24 men and women) aged between 50 and 65 years to evaluate the effects of 30-days resveratrol supplementation (500 mg/day) compared to energy restriction (1000 kcal/day) on the sympathetic nervous system activity (measured by plasma noradrenaline) and endothelium-dependent and independent vascular reactivity.

The participants of the study were hospital staff volunteers of both sexes in a 1:1 ratio with a normal clinical history, physical examination, and resting electrocardiogram. Women were postmenopausal (one year of natural amenorrhea). The exclusion criteria were smoking, hypertension (use of anti-hypertensive medication or diastolic blood pressure ≥90 mmHg), dyslipidemia (use of lipid-lowering medication, serum triglyceride concentrations ≥150 mg/dL, or total cholesterol ≥240 mg/dL), fasting glucose ≥110 mg/dL or use of hypoglycemic medication, hormone replacement therapy, premenopausal women, and non-adherence to diet and RSV supplementation (i.e., more than four days with energy intake over 1000 kcal, and more than two returning capsules of RSV, respectively). Other exclusion criteria were any previous self-reported history of chronic renal failure (serum creatinine ≥2 mg/dL), liver failure, or endocrine, hematological, respiratory, or metabolic clinically significant findings.

The clinical parameters analyzed were age, weight, body mass index (BMI), waist circumference, blood pressure, and heart rate. Biochemical parameters included serum concentrations of triglycerides, total cholesterol, high-density lipoprotein cholesterol (HDL-c), low-density lipoprotein cholesterol (LDL-c), apolipoprotein A–I (apoA–I), apolipoprotein B (apoB), lipoprotein (a) [Lp(a)], non-esterified fatty acids (NEFA), glucose, insulin, homeostasis model assessment for insulin resistance (HOMA-IR), high-sensitivity C-reactive protein (hsCRP), and noradrenaline (NA). Vascular reactivity parameters included baseline artery diameter and endothelium-dependent and independent dilation are described below.

After a 15-day washout (without the use of any medications or supplements), participants underwent a standardized interview, blood sample collection, anthropometric assessment (weight, BMI, and waist circumference), blood pressure and heart rate measurement, and vascular reactivity test. These tests were repeated at the end of the study. After randomization, 24 participants of the resveratrol (RSV) group received the supplement (250 mg RSV/capsule twice a day), whilst the 24 other participants comprised the energy restriction (ER) group. A standard hypocaloric diet with 1000 kcal/day was prescribed for the ER group, which corresponded to an approximate daily energy consumption reduction of 50%. Daily food diaries were used to analyze the adherence of the proposed interventions. Subjects were instructed to write down all the food intake day-by-day and to not exceed 1000 kcal/day.

The study was conducted at the Heart Institute (InCor) of the University of São Paulo Medical School, São Paulo, Brazil. All subjects provided written informed consent. The research protocol was approved by the Ethics Committee of the University of São Paulo Medical School Hospital (CAAE:00788012.8.0000.0068) and was in line with the Declaration of Helsinki (Trial registration: www.ClinicalTrials.gov, accessed on 13 May 2021; identifier: NCT01668836).

### 4.2. Biochemical Analyses

Venous blood was drawn after 12 h of fasting and put in tubes with and without an anticoagulant and then centrifuged for 20 min at 1800 g (Eppendorf, Hamburg, Germany) for plasma and serum separation, respectively.

Serum total cholesterol, triglycerides, HDL-c, and glucose were obtained by commercial colorimetric-enzymatic methods. LDL-c was calculated using the Friedwald equation. Measurements were performed using Dimension RxL equipment (Siemens Healthcare Diagnostic Inc., Newark, DE, USA) with dedicated reagents (Dimension^®^ Flex Reagent Cartridge). Lp(a), apoA-I, apoB, and hsCRP were obtained by immunonephelometry using dedicated reagents (Siemens N Latex^®^,Erlagen, Germany) for BN-II equipment from Siemens Healthcare (Marburg, Hessen, Germany). Insulin was analyzed by the chemiluminescence assay using automated equipment (Immulite 2000^®^ Insulin; Siemens Healthcare Diagnostic Inc., Newark, DE, USA). Serum NEFA was determined by a colorimetric kit from Randox Laboratories Ltd. (Crumlin, County Antrim, UK). Plasma NA was obtained through reversed-phase, ion-pair high-performance liquid chromatography (HPLC) coupled with electrochemical detection, following extraction by alumina adsorption according to a method previously described [56].

### 4.3. Vascular Reactivity Test

Endothelium-dependent flow-mediated vasodilation (ed-FMD) and endothelium-independent vasodilation (responses to 5 mg of sublingual isosorbide dinitrate—ei-NMD) were assessed under current guidelines [57]. Brachial artery diameters were assessed in the left arm in the recumbent position, after 10-min rest in a room kept at 20 °C to 25 °C, using a 7.5-MHz linear-array vascular ultrasound transducer and an Apogee 800 Plus ultrasound system (ATL Ultrasound, Bothell, WA, USA). Blood pressure and heart rate were monitored with an automated sphygmomanometer. Vessel diameter was measured in the longitudinal section where the lumen-intima was viewed from the anterior to the posterior wall by software that measures a segment of the artery and calculates an average. Reactive hyperemia was induced by the inflation of a tourniquet around the forearm to 250 mm Hg and deflated after 5 min. After 10 min, ei-NMD was performed. Endothelium-dependent and independent vasodilation were calculated as the percentage change in brachial artery diameter ratio after reactive hyperemia or nitrate to baseline diameter. All tests were performed and analyzed by a single dedicated ultrasonographer and according to recommendations of the International Brachial Artery Reactivity Task Force for endothelial function studies [57].

### 4.4. Resveratrol Purity and Formulation Analysis

The RSV administered to participants was obtained from a compounding pharmacy (Buenos Ayres Pharmacy, São Paulo, Brazil). The purity of the product supplied was analyzed by capillary electrophoresis using a Proteome Lab PA800 (Beckman Coulter, Fullerton, CA, USA) at the Laboratory of Capillary Chromatography and Electrophoresis at the Chemistry Institute of the University of São Paulo. Samples of the manipulated capsules and the standards of RSV were performed in triplicate, and areas under the peak were compared. The purity of RSV was 87% ± 1.1% on average (coefficient of variation: 1.2%).

### 4.5. Statistical Analysis

The sample size calculation was made by the difference between the plasma NA levels, before and after interventions. The difference between the means for RSV and ER groups was 100 pg/dL with a standard deviation of 100 pg/dL. The test power was β = 0.80 and α = 0.05. The estimated number of the sample was 24 subjects for each group. We also hypothesized that this reduction in NA plasma levels would be associated with an increase of 40% in FMD. Participants were randomly assigned in a 1:1 ratio with the use of computer-generated random numbers to receive either RSV or ER. The Kolmogorov–Smirnov test was used to assess variable distribution. Pre- and post-intervention variables were summarized with the use of descriptive statistics. Continuous parametric variables were summarized as the mean and standard deviation (SD) as well as the non-parametric, which were described as the median and interquartile range (IQR). Paired-sample *t*-test and Wilcoxon test were used for pre- and post-intervention comparisons. Independent *t*-test and Mann–Whitney U test were used for comparisons of intergroup changes. Correlations between changes (Δ, i.e., post-intervention—baseline) of variables with plasma noradrenaline and ed-FMD and ei-NMD were assessed by Pearson’s and Spearman’s correlations. Further associations of changes of total cholesterol, triglycerides, glucose, waist circumference, noradrenaline, ed-FMD, and ei-NMD were assessed by linear regressions, with the last three parameters being the dependent variables and the rest being predictor variables. Finally, logistic regression was applied to verify the independence of ei-NMD association with plasma NA. The level of significance was set at *p* < 0.05. The software used for statistical analysis was SPSS version 20.

## Figures and Tables

**Table 1 molecules-26-03168-t001:** Baseline and post-intervention characteristics of the participants.

	Resveratrol		*p*	Energy Restriction		*p*
	Baseline	30 days		Baseline	30 days	
	*n* = 24	*n* = 24		*n* = 24	*n* = 24	
Variables	Mean (SD)	Mean (SD)		Mean (SD)	Mean (SD)	
Age, *y*	58.5 (3.4)		-	58.6 (3.6)		-
Body mass index, *kg*/*m*^2^	27.6 (4.2)	27.8 (4.4)	0.370	25.8 (3.2)	25.5 (3.2)	0.083
Waist circumference, *cm*	96.8 (12.1)	96.9 (11.4)	0.457	94.2 (7.5)	91.8 (7.1)	**0.011**
Hear rate, *bpm*	64.6 (8.5)	65.7 (8.2)	0.269	62.5 (9.6)	62.3 (10.5)	0.902
Systolic blood pressure, *mmHg*	131.5 (15.5)	129.0 (15.4)	0.660	129.7 (15.6)	124.2 (12.8)	0.109
Diastolic blood pressure, *mmHg*	81.2 (10.8)	82.0 (9.2)	0.612	82.9 (11.0)	79.4 (9.9)	0.070
*Biochemical characteristics*						
Total cholesterol, *mg*/*dL*	207.71 (33.43)	217.58 (44.99)	**0.030**	216.21 (44.28)	202.67 (39.73)	**0.007**
HDL-c, *mg*/*dL*	49.17 (13.85)	48.17 (13.73)	0.260	55.33 (18.52)	52.00 (16.55)	**0.008**
LDL-c, *mg*/*dL*	132.33 (26.68)	139.54 (40.53)	0.089	138.67 (36.81)	130.29 (33.37)	**0.031**
Triglycerides, *mg*/*dL*	124.04 (66.39)	148.63 (93.00)	0.075	111.21 (63.09)	102.00 (61.01)	0.233
Apoliproprotein A-I, *g*/*L*	1.45 (0.21)	1.44 (0.24)	0.648	1.53 (0.27)	1.43 (0.28)	**0.011**
Apolipoprotein B, *g*/*L*	0.98 (0.24)	1.04 (0.27)	**0.034**	1.01 (0.26)	0.95 (0.26)	0.057
Lp(a), *mg*/*dL* *	10.87 (3.97–19.57)	9.73 (3.06–16.40)	0.688	11.95 (7.43–30.07)	14.55 (8.98–27.47)	0.463
Glucose, *mg*/*dL*	96.04 (13.67)	98.58 (14.26)	0.153	93.63 (10.68)	90.54 (5.89)	0.119
Insulin, *µUI*/*mL*	8.23 (5.48)	8.47 (5.42)	0.593	6.71 (4.37)	6.02 (3.14)	0.428
HOMA-IR	2.06 (1.63)	2.17 (1.75)	0.422	1.59 (1.25)	1.38 (0.75)	0.307
NEFA, *mEq*/*dL*	0.25 (0.15)	0.24 (0.10)	0.732	0.30 (0.37)	0.20 (0.17)	0.222
hs-CRP, *mg*/*L* *	1.81 (0.94–2.34)	1.46 (0.75–1.97)	0.512	1.36 (0.94–1.92)	1.18 (0.85–1.53)	0.474
Noradrenaline, *pg*/*dL*	269.33 (119.82)	223.58 (93.75)	0.065	347.50 (160.16)	253.64 (171.08)	**0.008**
*Vascular reactivity*						
Baseline artery diameter, *mm*	4.31 (0.65)	4.29 (0.64)	0.776	4.42 (0.91)	4.43 (0.98)	0.833
ed-FMD, *%*	3.26 (9.68)	3.06 (5.04)	0.940	3.41 (3.47)	1.97 (6.40)	0.395
ei-NMD, %	17.25 (8.21)	16.71 (6.89)	0.785	17.06 (6.14)	19.0 (8.81)	0.151

Data are presented in mean (standard deviation) or median (interquartile range) depending on variable’s distribution. * Wilcoxon’s test was applied for non-parametric variables. ed-FMD—endothelium-dependent flow-mediated vasodilation. ei-NMD—endothelium-independent nitrate-mediated vasodilation.

**Table 2 molecules-26-03168-t002:** Correlations between changes (Δ) of plasma noradrenaline, vascular reactivity, and cardiometabolic parameters.

Variables		Noradrenaline	WC	SBP	TC	HDL-c	LDL-c	TG *	ApoA-I	ApoB	NEFA	Glucose	HOMA-IR
*All groups (n* = *48)*													
Noradrenaline	*r*	1.000	0.300	0.130	0.443	0.263	0.263	0.438	0.467	0.318	−0.288	0.237	0.208
	*p*	-	0.060	0.405	**0.002**	0.077	0.078	**0.002**	**0.001**	**0.032**	0.058	0.112	0.166
ed-FMD	*r*	0.129	0.042	0.193	0.195	0.255	0.120	−0.140	0.018	0.150	−0.025	−0.127	−0.012
	*p*	0.446	0.821	0.253	0.249	0.127	0.481	0.445	0.916	0.374	0.885	0.454	0.946
ei-NMD	*r*	0.338	−0.056	−0.060	0.174	0.117	0.101	0.171	0.225	0.159	−0.327	0.041	0.244
	*p*	**0.041**	0.766	0.723	0.303	0.491	0.551	0.310	0.180	0.347	0.055	0.808	0.146
*Resveratrol group (n* = *24)*													
Noradrenaline	*r*	1.000	0.514	0.006	0.377	0.399	0.128	0.494	0.508	0.218	0.020	0.420	0.166
	*p*	-	**0.020**	0.980	0.069	0.053	0.551	**0.014**	**0.011**	0.306	0.931	**0.041**	0.438
ed-FMD	*r*	0.006	0.125	0.180	0.211	0.373	0.171	0.135	−0.112	0.265	−0.049	−0.048	0.326
	*p*	0.980	0.658	0.461	0.387	0.116	0.483	0.581	0.647	0.272	0.852	0.845	0.174
ei-NMD	*r*	0.336	−0.122	−0.132	0.389	0.354	0.157	0.435	0.469	0.307	−0.012	0.390	0.561
	*p*	0.160	0.664	0.589	0.100	0.137	0.522	0.063	**0.043**	0.200	0.963	0.099	**0.012**
*Energy restriction group (n* = *24)*													
Noradrenaline	*r*	1.000	0.028	0.158	0.449	0.134	0.324	0.338	0.403	0.314	−0.455	0.024	0.186
	*p*	-	0.907	0.482	**0.036**	0.552	0.141	0.124	0.063	0.155	**0.033**	0.917	0.408
ed-FMD	*r*	0.297	−0.536	0.225	0.152	0.147	−0.046	0.234	0.179	−0.014	−0.061	−0.338	−0.454
	*p*	0.231	**0.033**	0.368	0.547	0.559	0.855	0.350	0.478	0.955	0.810	0.170	0.058
ei-NMD	*r*	0.456	−0.061	0.042	0.190	0.040	0.212	−0.077	0.089	0.248	−0.468	−0.184	0.118
	*p*	0.057	0.822	0.869	0.451	0.875	0.398	0.763	0.724	0.322	0.050	0.466	0.640

* Spearman’s correlation was applied for non-parametric variables. WC—waist circumference. SBP—systolic blood pressure. TC—total cholesterol. TG—triglycerides. NEFA—non-esterified fatty acids. HOMA-IR—homeostasis model assessment for insulin resistance. ed-FMD—endothelium-dependent flow-mediated vasodilation. ei-NMD—endothelium-independent nitrate-mediated vasodilation.

**Table 3 molecules-26-03168-t003:** Associations of variable changes after resveratrol and energy restriction treatments (Δ) with plasma noradrenaline.

	Simple Model				Multivariate Model *			
	Noradrenaline				Noradrenaline			
Predictor Variables (Δ)	β	SE	OR	(95%CI)	β	SE	OR	(95%CI)
ei-NMD, *%*	0.189	0.074	1.208	**(1.044; 1.398)**	0.258	0.121	1.294	**(1.021; 1.640)**
Triglycerides, *mg*/*dL*	0.022	0.010	1.022	**(1.003; 1.042)**	0.027	0.018	1.027	(0.991; 1.065)
Total cholesterol, *mg*/*dL*	0.036	0.016	1.037	**(1.005; 1.069)**	0.026	0.025	1.026	(0.977; 1.078)
Waist circumference, *cm*	0.011	0.023	1.011	(0.966; 1.058)	0.016	0.031	1.016	(0.955; 1.081)

β—coefficient of logistic regression. 95%CI—95% confidence interval. OR—odds ratio. * Endothelium-independent nitrate-mediated vasodilation (ei-NMD), triglycerides, total cholesterol, and waist circumference were the predictor variables of the model.

## Data Availability

No new data were created or analyzed in this study. Data sharing does not apply to this article.

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
