# Peer review of "Comparison of Resveratrol Supplementation and Energy Restriction Effects on Sympathetic Nervous System Activity and Vascular Reactivity: A Randomized Clinical Trial"

_molecules, 2021, doi:10.3390/molecules26113168_

Round 1
Reviewer 1 Report
In conclusion, the results of the study have shown that short-term RSV supplementation does no improve cardiometabolic risk factors, sympathetic activity and endothelial function, whereas short-term ER decreases plasma NA and waist circumference, as well as improves blood lipids, but does not modify endothelial function in overweight healthy adults.
In this study, authors examined that the effect of resveratrol and energy restriction on plasma noradrenaline and endothelial functions in non-obese patient.
In this study, authors examined that the effect of resveratrol and energy restriction on plasma noradrenaline and endothelial functions in non-obese patient. However, the effect of both on vascular endothelial function was not observed. The reason may be that the observation period was short, and the background of the patients was also different from the previous reports.
In table1, it has been suggested that ER may suppress the activity of the sympathetic nervous system. However, the plasma NA concentration at the baseline appears to be different between groups. If there is a significant difference, the validity of the subsequent analysis is questionable in Table 2.
In the discussion, I mentioned the possibility of improving insulin resistance by RSV and ER, but I don't think it is justified as far as the HOMA-IR of patients are concerned. Overall, the consideration is too overstated.
Reviewer 2 Report
Authors describe in the manuscript the outcomes of a study with healthy patients taking resveratrol or energy reduced diet. Several markers were studied, such as plasma noradrenaline levels, cholesterol, triglicerides etc. and endothelium vasodialtion.
Although the manuscript seems to be composed with care and dedication, there are several details that require further attention and/or correction.
As of English language, the overall level of the manuscript is satisfactory and, with the exception of a couple of details, i feel there is no need to significantly improve the grammar of the article.
Minor coments:
- Sentence 23-24 in the abstract requires reformulation. As in the present form, it is quite confusing.
- Sentence 62-64 is irrelevant. Authors should eliminate or reformulate it
- Sentence 128: “aimed to” between “study” and “evaluated” would be a more precise term for the manuscript
- Sentence 354: Country of origin of the Apogee system is missing
Major comments:
- Result section is quite messy. Authors should include subsections in which they can in detail describe the results of each treatment. Perhaps a subsection for each table included would improve the manuscript quality and readability
- Furthermore, the results section requires a better result description. In my opinion, the actual description is quite confusing for the reader - an overall richer result description is necessary
- How did the authors assess that trial individuals were complying with the study conditions? - this is in my opinion the weakest point in the trial since, as i understood, individuals were healthy workers which could easily fail to comply with the study requirements in any social event (birthday celebrations, weekend alcohol consumption etc.)
- In materials and methods:
- Authors should include a criteria compliance declaration or confirmation
- Biochemical analysis subsection needs thorough improving:
- Each method used should be described in detail, with the procedures used, reagents and chemicals used, origin etc.
- Especially for HPLC, the use of one sentence is highly insufficient. Detailed information of this methodology is required and needed prior to publication
- Perhaps an additional subsection for each method would be appropriate
- In 4.3., in sentence 351 writes that vasodilation experiments were performed under current guidelines but fail to precise which these guidelines are. At least a reference is here missing
In spite of the comments referred, after the correction and complementation of the manuscript, I recommend the acceptance of this article for publication in Molecules journal.
Reviewer 3 Report
The aim of the study was to compare the effects of resveratrol supplementation and energy restriction on sympathetic nervous system activity and vascular reactivity.
I have some questions and comments:
1. In energy restriction (ER) group you found after intervention reduction of concentrations of both NA and total cholesterol (Table 1). In resveratrol (RSV) group was increased total cholesterol and non-significantly decreased NA. Was really overall correlation between noradrenaline (NA) and total cholesterol concentrations?
2. In ER group was observed significant reduction of NA concentrations 30 days after intervention. The reduction of NA concentrations was also in RSV group but these changes were not statistically significant. However, in ER group was much higher mean NA concentration at the baseline in comparison to RSV group (347.5 pg/dL vs 267.3 pg/dL). Could you comment these data?
3. You wrote (lines 223-224): “It is important to note that plasma triglycerides were positively correlated with NA in the RSV group …” In case of positive correlation variables move in the same direction.
In you study was mean value of triglycerides concentration after RSV intervention higher in comparison to baseline value, mean value of NA was lower. Reflects this discrepancy inter-individual variability of these parameters at participants of RSV group?
4. Sympathetic nervous system activity you determined only through changes in NA concentrations. You described this fact also as a limitation of study. Is from this point correct the statement in title that you compared the effects of resveratrol supplementation and energy restriction on sympathetic nervous system activity?
5. In your study you did not determine plasma concentrations of NA and other biomarkers. Based also on description of methods you did not use plasma but serum.
Round 2
Reviewer 1 Report
Authors mentions that the objection of this study is to clarify the effects of resveratrol supplementation and energy restriction on vascular function and the SNS activity in lane 82. However, the authors suggested that only a relationship between SNS and vascular function. This study maybe lacks power to clarify the resveratrol supplementation and energy restriction on vascular function. I think that the title and discussion are not based on the results.
Author Response
Dear Reviewer. Thanks again for your time in reviewing our manuscript. You could be right in relation to flow-mediated vasodilation variables, because our sample size calculation was made for changes in NA plasma levels, and we hypothesized that this amount of reduction in NA plasma levels would be associated with an increase of 40% in flow-mediated vasodilation. If we also do a sample size calculation of 40% increase in flow vasodilation using the values observed in previous studies, we will see nearly the same sample size of 24 subjects we used (Am J Clin Nutr 2017;105:23-31 and High Blood Press Cardiovasc Prev 2019;26(4):305-319). Nevertheless, we did not find differences in flow-mediated variables after interventions, and I think this happened because we studied a homogeneous group of normal subjects. But we did find a moderate correlation between delta (pos- minus pre-intervention) values of NA and endothelium-independent blood flow. We think this latter finding of delta values is more important than pre and post-intervention values.
By the way, we included our sample size calculation in the Statistical Analysis section of Methods.
Reviewer 3 Report
Authors answered questions and comments that I have addressed to them by revision of the previous version of manuscript and included some corresponding changes into revised manuscript.
Author Response
Dear Reviewer. Thanks again for your time in reviewing our manuscript.